# Training LLM Agents to Empower Humans

## Abstract

Assistive agents should not only take actions on behalf of a human, but also step out of the way and cede control when there are important decisions to be made. However, current methods for building assistive agents, whether via mimicking expert humans or via RL finetuning on an inferred reward, often encourage agents to complete tasks on their own rather than truly *assisting* the human attain their objectives. Additionally, these methods often require costly explicit human feedback to provide a training signal. We propose a new approach to tuning assistive language models based on maximizing the human's *empowerment*, their ability to effect desired changes in the environment. Our empowerment-maximizing method, `Empower`, only requires offline text data, providing a self-supervised method for fine-tuning language models to better assist humans. To study the efficacy of our approach, we conducted an 18-person user study comparing our empowerment assistant with a strong baseline. Participants preferred our assistant 78% of the time ($p = 0.015$), with a 31% higher acceptance rate and 38% fewer suggestions. Additionally, we introduce a new environment for evaluating multi-turn code assistance using simulated humans. Using this environment, we show that agents trained with `Empower` increase the success rate of a simulated human programmer on challenging coding questions by an average of 192% over an SFT baseline. With this empowerment objective, we provide a framework for useful aligned AI agents at scale using only offline data without the need for any additional human feedback or verifiable rewards.

Website and code: https://anonymous.4open.science/r/codegen-384F/

## 1 Introduction

Software developers today face a challenge when using LLM coding agents: code suggestions start out helpful but then start implementing the wrong functions. Often an assistant will suggest a large block of code, the user accepts it, and then they have to spend time fixing the one part it got wrong, such as an incorrect assumption. How can we develop coding assistants that still produce helpful generations, but also know to stop their generations at critical junctures? While this problem is especially salient for coding assistants (the focus of this paper), such problems are likely to recur in applications from assistive robotics to interacting with autonomous web agents (Chen et al., 2021; Trivedi et al., 2024).

Optimizing for helpfulness is challenging. Gathering explicit human labels is expensive and time-consuming. Additionally, it is unclear how this sort of helpfulness can be learned from traces of a human expert — the problem is not that generations are unrealistic, but rather that they may be solving a problem that is different from what the user intends. One approach to this problem is for agents to ask clarifying questions to better infer the intentions of human users. However, this style of assistance requires interrupting the user, possibly impeding their flow and making the interaction feel burdensome. In many situations, it is desirable to have an assistant which does not rely on querying the user.

The key insight in this paper is to train assistive agents to *empower* human users, a training method that does not require that the agents know the human's underlying intention. Intuitively, empowerment refers to an agent's ability to effect changes in the environment. Rather than asking a human for explicit feedback, the LLM will automatically assess the usefulness of its actions by estimating whether they enable a human to solve more tasks more quickly. In coding contexts, empowerment might correspond to implementing helper functions, writing boilerplate, or wrapping up lines of code.

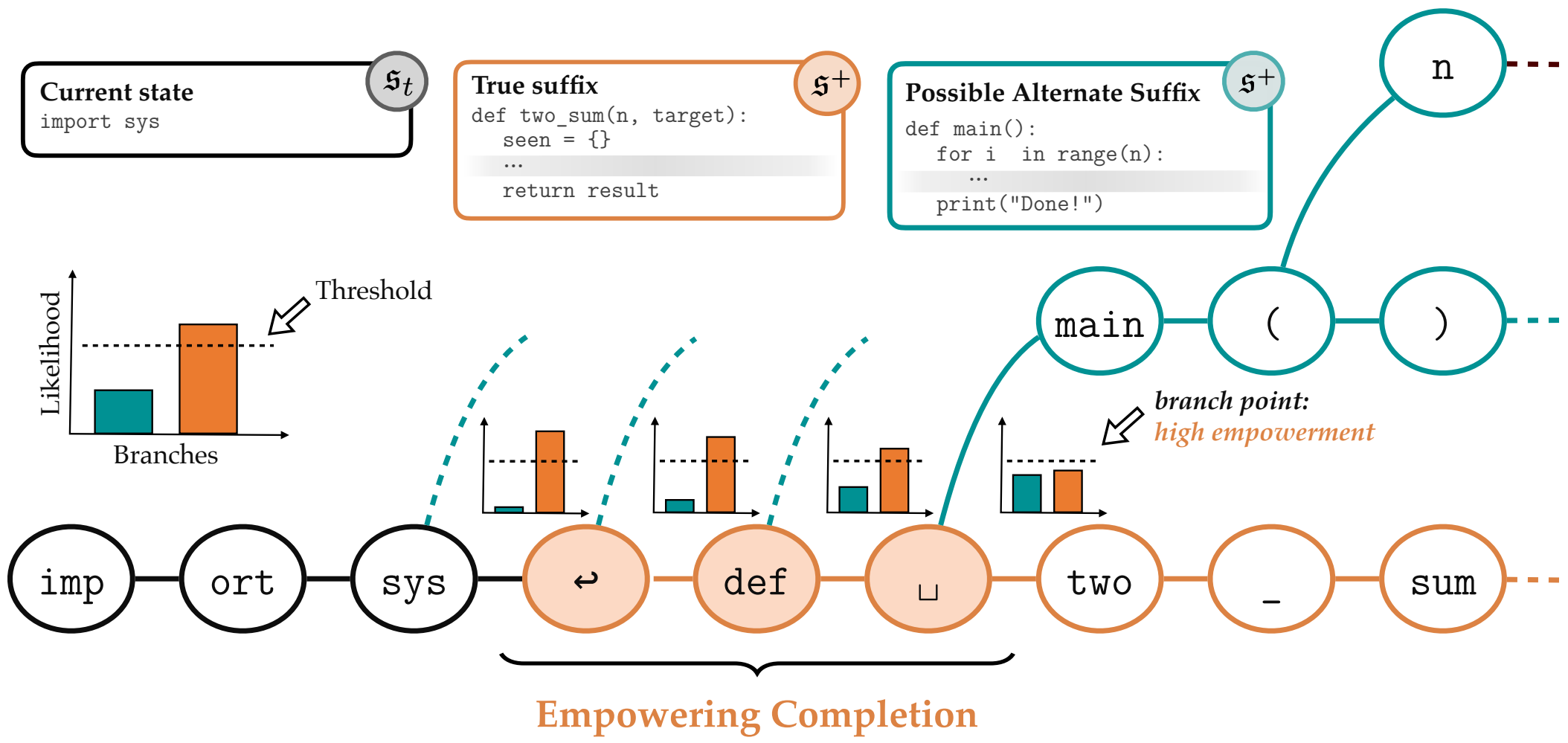

Figure 1: **Training assistive agents via `Empower`.** An LLM generates the cumulative likelihood of the suffix, shown below each token. Empowering completions are selected as the longest suffix where the cumulative likelihood is greater than a threshold. This trains the assistant to complete text up to a decision point. Then, the human will have more choices about where to take the program, so their next action is *empowered*.

Mathematically, empowerment is defined by a mutual information that measures the degree of control that an agent's actions exert on states that occur in the future (Klyubin et al., 2005). The assistive setting, where there are two agents (an assistant and a human), requires a more nuanced definition of empowerment: we aim to empower the *human* agent, enabling the human user's actions to exert a larger influence over future outcomes (Du et al., 2020; Myers et al., 2024). An agent that maximizes the human's empowerment will help them reach goals more effectively, without assuming any prior reward structure. Similar empowerment objectives are used in psychology to explain certain facets of human learning (Gopnik, 2024).

We use code generation as a context for studying empowerment-maximizing assistants because it is one of the few real-world applications today where humans regularly interact with assistive agents (e.g., Github Copilot). It is also an appealing starting point because preexisting datasets (Jain et al., 2024) allow us to measure the efficacy of empowerment maximization in a rigorous way.

**Contributions.** In this paper, we derive a practical and scalable algorithm for training LLM agents to maximize empowerment, and demonstrate its effectiveness in the code generation setting. Specifically, our work makes the following contributions:

1. **`Empower` method (Section 4.2).** We propose `Empower`, a method for aligning LLM agents to work with humans based on the objective of maximizing effective empowerment. Our method provides a proof-of-concept for training an LLM agent to maximize a human's empowerment.

2. **Simulated results (Section 5.2).** Experimental results with a Gemma-3-27B-it (Team et al., 2025) human on LiveCodeBench (Jain et al., 2024) show that our method leads to a higher Pass@1 without explicit human feedback. A Llama-3.1-8B-Instruct model trained with `Empower` over doubles the Pass@1 rate compared to the strongest baseline.

3. **User study (Section 5.3).** We demonstrate in a user study that human coders prefer our empowerment assistant over a baseline. Participants preferred our assistant in practice 78% of the time ($p = 0.015$), and accepted our assistant's suggestions 31% more often than the baseline's suggestions ($p = 0.0002$). Additionally, participants tended to delete 26% fewer characters they had accepted from our method than from the baseline ($p = 0.012$). This amounts to a stronger assistant that is more likely to suggest code the user will accept and actually use.

Taken together, our results demonstrate that LLM assistants can be trained without receiving feedback or interaction from humans by reasoning about how their actions might enable humans to complete more tasks more quickly.

## 2 RELATED WORK

Past work has studied empowerment in the context of intrinsic motivation and reinforcement learning.

**Empowerment.** Informally, empowerment quantifies the influence an agent's actions have over outcomes in the environment. Empowerment has traditionally been defined as the channel capacity between a sequence of actions and the following state (Abel et al., 2025; Klyubin et al., 2005; 2008; Salge et al., 2014). Empowerment objectives have been used in single-agent reinforcement learning to enable intrinsic motivation and exploration (Choi et al., 2021; de Abril and Kanai, 2018). More recently, empowerment has been explored for collaborative settings, where a robot assistant learns to maximize the empowerment of a human user. Du et al. (2020) propose the AvE algorithm which enables assistance by computing empowerment with random rollouts. Myers et al. (2024) adopt a modified objective, the effective empowerment, which can be learned with a scalable contrastive objective. However, these prior works typically use simple gridworld-like or video game-like environments. On larger web agent benchmarks, Song et al. (2025) found that the effective empowerment is highly correlated with LLM task performance. Their work focused on evaluating the correlation between effective empowerment and reward, rather than on optimizing LLM agents with an empowerment objective. To the best of our knowledge, ours is the first to apply this principle to train LLM agents, applying it at scale to a realistic coding task. This is enabled by our insight that LLM uncertainty can be used to identify key decision points, and we can empower people by helping them reach those points.

**Learning from Human Preferences.** Many methods attempt to align AI agents by updating the agent with human preference information. Christiano et al. (2017) used an online stream of human preferences to train a reward model concurrently with an actor-critic policy. This method (RLHF) was later adapted to align LLMs to human preferences, enabling conversational agents like InstructGPT (Dong et al., 2023; Ouyang et al., 2022; Stiennon et al., 2020). These methods often use PPO or related policy-gradient methods to fine-tune a pre-trained LLM (DeepSeek-AI et al., 2025; Schulman et al., 2017; Sutton and Barto, 2018). Learning and optimizing a reward model can be expensive or unstable, leading to alternative methods like DPO (Rafailov et al., 2024) and IPL (Hejna and Sadigh, 2023) that directly optimize the policy to match human preferences without an intermediate reward model. Training with human feedback faces key limitations: human values may be difficult to represent with reward functions (Casper et al., 2023), they may change over time (Carroll et al., 2024), and optimizing them may lead to misaligned behaviors like power seeking and manipulation (Williams et al., 2024). Our empowerment objective offers an alternative strategy for aligning LLMs which instead completes tasks for the human that are obvious and general, rather than aligning assistance to a set of preferences.

**Self-Supervision for LLMs.** There is existing precedent for having LLMs provide their own feedback, as in self-critiquing methods that are common in mathematical and logical reasoning applications (Madaan et al., 2024; Shinn et al., 2024; Yao et al., 2024) or methods that aim to leverage the model itself to provide a learning signal for self improvement (Huang et al., 2025; 2023; Pang et al., 2023; Yuan et al., 2024). In contrast, our work places the human back at the center: rather than optimizing an LLM to produce text that another LLM thinks is correct, we optimize an LLM to produce code that enables a human to solve more tasks more quickly.

**Assistive Agents.** One mathematical framework for assistive agents is the assistance game (Hadfield-Menell et al., 2016). The assistance game extends the standard Markov decision process (MDP) definition to include a human and a robot agent which interact in a shared environment to maximize a joint reward that is only known to the human. The agent must combine inference of the human rewards or goals with reinforcement learning to optimize the inferred objective (Carroll et al., 2024; 2019; Hadfield-Menell et al., 2017; Laidlaw et al., 2024). Methods that learn from human preferences (Christiano et al., 2017; Rafailov et al., 2024) can be seen as special cases of the assistance game where the human's actions only exist to provide information about the reward to the agent. Other works within the assistive setting focus on task completion with humans in the loop, where the robot assistants need to learn when to ask humans for help and what to ask them for (Mullen Jr and Manocha, 2024; Ramrakhya et al., 2025; Ren et al., 2023). Empowerment methods are a special case of assistive games, where the empowerment objective acts as a proxy for the

human's reward function (Myers et al., 2024). Our work builds on this foundation to study how empowerment methods might be scaled to align LLMs.

Our contribution is to connect empowerment to the problem of aligning LLM agents to human users, showing that the empowerment objective can provide a scalable self-supervised learning signal for an LLM agent aiding a human in an assistive setting.

## 3 PRELIMINARIES

We cast the problem of an LLM agent assisting a human as a Markov decision process (MDP). The state is the program text at the given timestep. At each state, the LLM agent action suggests a piece of text to append to the conversation. The human agent first chooses to `ACCEPT` or `REJECT` the suggestion, or `FINISH` writing the program. Then, unless they choose to `FINISH`, they will append some number of tokens.

**Notation.** In this assistance MDP, the human policy $\pi_{\mathbf{H}}$ selects an action $a^{\mathbf{H}} \in \mathcal{A}_{\mathbf{H}}$, and the LLM agent's policy $\pi_{\mathbf{R}}$ selects an action $a^{\mathbf{R}} \in \mathcal{A}_{\mathbf{R}}$ to complete the code snippet. Let $\mathcal{L}$ be the set of possible tokens and $\mathcal{L}^*$ be the set of all strings consisting of these tokens. The human's actions are $\mathcal{A}_{\mathbf{H}} = (\{\texttt{ACCEPT}, \texttt{REJECT}\} \times \mathcal{L}) \cup \{\texttt{FINISH}\}$, where `ACCEPT` appends the LLM agent's suggestion to the conversation followed by the human's own text, `REJECT` does not append the suggestion and only appends the human's text, and `FINISH` ends the episode. If the human does not choose `FINISH`, they will write one token. The LLM agent's actions are $\mathcal{A}_{\mathbf{R}} = \mathcal{L}^*$, where the agent suggests any sequence of tokens to append to the conversation. Note that $\ell_{i:j}$ indexes the tokens from $i$ to $j$ inclusive. We will use $s_t = \ell_{1:n}$ to indicate that the current state is $n$ tokens. The assistant's suggestion is $a_t^{\mathbf{R}} = \ell_{n+1:n+i}$, where $i$ is the number of tokens it proposed. If the human accepts the suggestion, then the human will write $\ell_{n+i+2}$. If the human rejects the suggestion, then they will write $\ell_{n+1}$.

The dynamics are then defined by the following transitions when $s_t = \ell_{1:n}$ and $a_t^{\mathbf{R}} = \ell_{n+1:n+i}$:

$$s_{t+1} = \begin{cases} \ell_{1:n+i+2} & \text{for } a_t^{\mathbf{H}} = (\texttt{ACCEPT}, \ell_{n+i+2}) \\ \ell_{1:n+1} & \text{for } a_t^{\mathbf{H}} = (\texttt{REJECT}, \ell_{n+i+1}) \\ \bot & \text{for } a_t^{\mathbf{H}} = \texttt{FINISH}. \end{cases}$$

We will define random variables $\mathfrak{s}_t$ and $\mathfrak{a}_t^{\mathbf{H}}$ to denote the state and human action at time step $t$. Similarly, in state $s_t = \ell_{1:n}$ if the human does not accept the assistant's suggestion let $\ell_{n+1}^{\mathbf{H}}$ represent the random variable of the next token that the human writes. Let $\ell^+$ be a random variable over possible future text. Additionally, $\hat{\pi}(\ell_{n+1:n+i} \mid \ell_{1:n})$ denotes a conditional probability distribution over possible completions. We will choose this to be a pre-trained LLM.

**Empowerment.** We will define empowerment using the mutual information $I(\cdot;\cdot)$ between two random variables. In a single-agent MDP, empowerment is defined by Klyubin et al. (2005) as the channel capacity between a sequence of $n$ actions and the resulting state $n$ steps into the future:

$$C\big(p(\mathfrak{s}_{t+n} \mid \mathfrak{a}_t^n, \mathfrak{s}_t)\big) \triangleq \max_{p(a_t^n \mid \mathfrak{s}_t)} I(\mathfrak{a}_t^n; \mathfrak{s}_{t+n} \mid \mathfrak{s}_t). \tag{1}$$

Informally, this objective states that empowerment is the maximal degree to which the next $n$ actions $\mathfrak{a}_t^n$ selected in the MDP can impact the resulting state $\mathfrak{s}_{t+n}$. This objective is intuitively appealing because it provides a mathematical way of quantifying whether an assistive agent's actions are useful, without knowing the humans reward function. However, it is challenging to use this objective in practice because *(i)* it involves optimizing over a sequence of actions, and *(ii)* mutual information is still non-trivial to compute in high-dimensional environments or over long horizons.

## 4 MAXIMIZING EMPOWERMENT OVER LANGUAGE

In this section we discuss how to train empowering assistants in the language domain. Empowerment is a useful objective for assistance because it helps people quickly reach states where they have many choices, so it takes broadly useful actions that are helpful for the most people. This leads to a more

**Algorithm 1:** Logit Threshold Empowerment (`Empower`)

**Input:** A text document $\ell_{1:N}$ with sampled state $\ell_{1:n}$
**Output:** Empowering suggestion $\ell_{n+1:n+i^*}$ for state $\ell_{1:n}$

```
1: for i ∈ {1...N} do                          ▷ Loop through the possible completion lengths
2:     Ĥ ← −log π(ℓ_{n+1:n+i} | ℓ_{1:n})       ▷ Compute the one-sample entropy
3:     if Ĥ > η then                           ▷ Check if estimated entropy exceeds threshold
4:         return ℓ_{n+1:n+i−1}                ▷ Return the last index that was below threshold
5: return ℓ_{n+1:N}                            ▷ Entropy is always within bounds, so return rest of program
```

natural type of assistance that does not make assumptions about, or even try to infer, the human's goal.

Section 4.1 introduces the core problem of constructing a training dataset for the assistant with only offline human data. Section 4.2 describes `Empower`, our practical method for choosing completions to train the assistant. We take the longest completion where the cumulative likelihood of the completion, as judged by an LLM, is above a certain threshold. Section 4.3 shows that, under certain assumptions, our method is computing an approximate upper bound on the empowerment of a completion. We train the LLM assistant to complete text that has a low empowerment — text that is predictable — so that the human does not have to write it. Instead, the human can focus on important design decisions, rather than boilerplate code.

### 4.1 Using Offline Data

Given an offline dataset of text written by the human, $\ell_{1:N} \sim \pi_{\mathbf{H}}$, the challenge is in choosing the best state-action pairs for finetuning the assistant. One approach would have an assistant model generate a proposed completion and then use some external feedback, such as whether the human would accept the suggestion, to score its quality. However, we do not assume access to human preference data, so the training signal must come from the text the human wrote. Therefore, we will train the assistant to output the same text that the human wrote in the dataset. This removes the need for `ACCEPT`/`REJECT` feedback, because the assistant proposes text that the human actually wrote, which we assume would be accepted. Formally, we train the assistant to output $\ell_{n+1:n+i}$ when it is given $\ell_{1:n}$ as the state. For each piece of text in the dataset, we sample a single state $\ell_{1:n}$ by uniformly sampling $t \in [1, N]$. The difficulty lies in choosing the appropriate length of completion to train on.

### 4.2 Our Algorithm: Empower

When the human writes boilerplate code, they have a low empowerment because their actions are easily predicted, so they carry little information about the future. To empower the human, an assistant should be trained to complete this predictable text so that the human does not have to. Our insight is that we can use an LLM, $\hat{\pi}$, to estimate how likely a completion is. We therefore propose the following algorithm to choose completions to train our assistant on:

$$i^* = \arg\max_i \big\{ i : -\log \hat{\pi}(\ell_{n+1:n+i} \mid \ell_{1:n}) < \eta \big\}. \tag{2}$$

This optimization chooses the largest completion length, $i$, such that the negative log likelihood of that completion as judged by an LLM is below a threshold $\eta$ which we choose. This can equivalently be viewed as choosing the longest completion length, $i$, where the cumulative likelihood of the completion is greater than $2^{-\eta}$. We write the optimization with a negative log likelihood to highlight that it is a one-sample estimate of the entropy. This mathematically connects our method to empowerment, which we will explain further in Section 4.3.

During training, we first sample a program from an offline dataset, then sample a prefix to that program which becomes the state $\ell_{1:n}$. Any suffix is a possible completion. We train on the suffix $\ell_{n+1:n+i^*}$ chosen by Eq. (2). Intuitively, we are training the assistant on obvious completions — those that the LLM thinks are likely — thereby leaving the human to write more impactful text in the future. We summarize our method in Algorithm 1, and show an illustration in Fig. 1.

### 4.3 MATHEMATICAL CONNECTIONS WITH EFFECTIVE EMPOWERMENT

Under some assumptions, our algorithm can be viewed as training the assistant to suggest text that would have a low empowerment for the human to write. We use the *effective empowerment* objective (Myers et al., 2024), which provides a computationally-tractable alternative to the canonical empowerment objective (Klyubin et al., 2005) (see Eq. (1)). Effective empowerment is defined with respect to a specific policy, $\pi_{\mathbf{H}}$, and a future state $\ell^+$. We define the effective empowerment at a state $\ell_{1:n}$ as:

$$\mathcal{E}(\pi_{\mathbf{H}}, \ell_{1:n}) \triangleq I(\ell_{n+1}^{\mathbf{H}}; \ell^+ \mid \ell_{1:n}). \tag{3}$$

This is the same objective introduced in (Myers et al., 2024), but with $\gamma = 0$. This objective measures the impact that the human's action, $\ell_{n+1}^{\mathbf{H}}$, has on their future state. We can upper-bound the mutual information with an entropy:

$$I(\ell_{n+1}^{\mathbf{H}}; \ell^+ \mid \ell_{1:n}) = H(\ell_{n+1}^{\mathbf{H}} \mid \ell_{1:n}) - H(\ell_{n+1}^{\mathbf{H}} \mid \ell^+, \ell_{1:n}) \tag{4}$$

$$\leq H(\ell_{n+1}^{\mathbf{H}} \mid \ell_{1:n}). \tag{5}$$

If we can estimate $H(\ell_{n+1}^{\mathbf{H}} \mid \ell_{1:n})$, we can estimate an upper bound for single-action empowerment. Computing this entropy exactly requires knowing the true human policy, $\pi_{\mathbf{H}}(\ell_{n+1}^{\mathbf{H}} \mid \ell_{1:n})$, which we do not have access to. Instead, we assume access to another likelihood estimator, $\hat{\pi}(\ell_{n+1}^{\mathbf{H}} \mid \ell_{1:n})$, which can approximate the human's marginal likelihood of any action at a given state. Then we can approximate the human's marginal entropy $H(\ell_{n+1}^{\mathbf{H}} \mid \ell_{1:n})$ by sampling an action from the human and using a one-sample monte carlo estimate:

$$\hat{H}(\ell_{n+1}^{\mathbf{H}} \mid \ell_{1:n}) \approx -\log \hat{\pi}(\ell_{n+1}^{\mathbf{H}} \mid \ell_{1:n}). \tag{6}$$

In practice, we choose our human entropy estimator $\hat{\pi}$ to be a pre-trained LLM. Our estimated upper bound on the empowerment becomes:

$$\mathcal{E}(\pi_{\mathbf{H}}, \ell_{1:n}) \lessapprox -\log \hat{\pi}(\ell_{n+1}^{\mathbf{H}} \mid \ell_{1:n}). \tag{7}$$

While this is a rough approximation of the entropy, it works well in practice for the purpose of choosing empowering completions, and is simple to implement. This approximation does not capture the human's ability to control the environment, which would be measured in the $H(\ell_{n+1}^{\mathbf{H}} \mid \ell^+, \ell_{1:n})$ term of Eq. (4). In a textual environment such as ours, the human's control is complete because they fully determine what is included in the final program. In fact, in our exact MDP described in Section 3, this bound is an equality because the human's action is to append a single token onto the state, so knowing $\ell^+$ and $\ell_{1:n}$ we can exactly deduce $\ell_{n+1}^{\mathbf{H}} = \ell_{n+1}^+$. Therefore, $H(\ell_{n+1}^{\mathbf{H}} \mid \ell^+, \ell_{1:n}) = 0$ and

$$\mathcal{E}(\pi_{\mathbf{H}}, \ell_{1:n}) \approx -\log \hat{\pi}(\ell_{n+1}^{\mathbf{H}} \mid \ell_{1:n}). \tag{8}$$

Although our approximate bound does not estimate control, this is a reasonable reduction for many textual environments. Under these assumptions, the algorithm described by Eq. (2) can be seen as training an assistant to complete text which is *predictable*, and therefore would not be empowering for the human to write.

## 5 EXPERIMENTS: CODE GENERATION

Our experiments apply the empowerment framework discussed in Section 4 to the task of code generation. In Section 5.1, we describe our experiment setup. We then evaluate our method in a novel simulated setup using LiveCodeBench (Section 5.2), after which we validate our findings in the real world by running an 18-person double-blinded human study (Section 5.3).

### 5.1 EXPERIMENT SETUP

**Datasets.** We train all models and methods using a dataset of 4,138 unique questions from Codeforces[1], each of which is paired with one attempted solution by Gemma-3-27B-it (Team et al., 2025). We do not filter the dataset for success on the testcases.

[1] `https://huggingface.co/datasets/MatrixStudio/Codeforces-Python-Submissions`

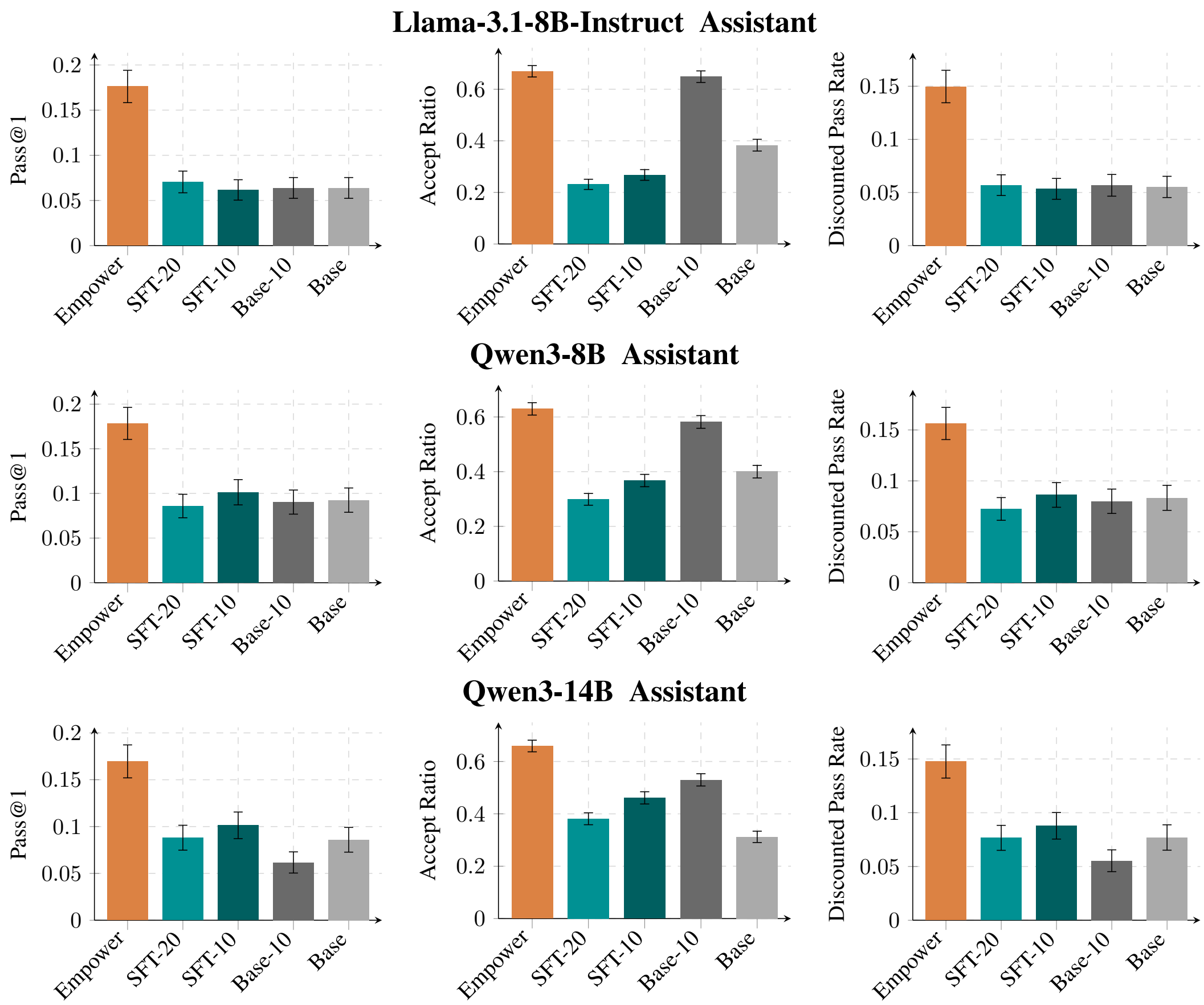

Figure 2: **Assistant results with Gemma-3-27B-it as the human model.** We evaluate on 554 LiveCodeBench problems. We find `Empower` to outperform all baselines in terms of pass@1 and DPR. Error bars show standard errors.

**Models.** We use Llama-3.1-8B-Instruct (Grattafiori et al., 2024), Qwen3-8B (Yang et al., 2025), and Qwen3-14B (Yang et al., 2025) as assistant models. For the simulated setting, we use Gemma-3-27B-it (Team et al., 2025) as the human model. The prompts we use are provided in Section D. We use all models with their default sampling parameters.

**Baselines.** We compare against both trained and untrained baselines. **(1) SFT-N** finetunes the assistant on the next $N$ tokens that the human wrote in a particular state, followed by a stop token. This should teach the model to output correct suggestions which are not too long, so that they do not make too many assumptions about what the human is trying to do. We evaluate SFT-10 and SFT-20. **(2) SFT-RAND** trains on random human completions between 1 and 30 tokens long to avoid biasing too much towards a specific completion length. **(3) Base** is simply the base assistant model without any training or restrictions on top. **(4) Base-N** is the same as Base, but we cap the suggestion length at N tokens. We include this baseline since we hypothesize that shorter completion lengths are more likely to be accepted. We evaluate Base-10 and Base-20.

**Our Method.** Our method, `Empower`, trains on completions returned from Algorithm 1, which we run on all completions in the training dataset before the start of training. We use the untrained base assistant model as our likelihood estimator, $\hat{\pi}$. Crucially, we do not provide the likelihood model access to the relevant Codeforces problem, only the text in the state (i.e. the completion tokens written so far).

### 5.2 Evaluating empowerment in a simulated setup

To evaluate the empowerment assistant with a simulated human, we adopt the MDP structure described in Section 3 where the assistant proposes suggested code completions which the human

may accept or reject, and then append their own code. We limit the human action size to $K_{\mathbf{H}} = 10$ tokens and the number of rounds of human and assistant actions per problem to 50. We evaluate on LiveCodeBench (Jain et al., 2024), a benchmark of competitive programming problems that is regularly updated. We restrict the benchmark to problems from release #6 to avoid contamination.

**Evaluation Metrics.** To evaluate the performance of `Empower` compared to the baselines, we propose the following three different evaluation metrics. **(1) Pass@1** measures the success rate of the generated code snippets by evaluating them on the problem testcases, counting a success only if all of the testcases pass. The results are averaged across all problems in the dataset. **(2) Acceptance rate** provides a measure of the human's preference for one assistant's suggestions over another's.

Our third metric is **(3) Discounted Pass Rate (DPR).** A higher acceptance rate is not always beneficial if the suggested completions are not more helpful. Occasionally an assistant will propose a completion which looks good, but actually introduces a bug or confuses the human, leading to a lower pass@1. Similarly, a lower acceptance rate can lead to a higher pass@1, making an assistant appear better even though the real gain in performance is from the human solving the problem on their own. Therefore, we introduce a new metric which we call the *Discounted Pass Rate* (DPR), which is a better measure of good assistance because it accounts for both the pass rate and the amount of text the human had to read and write to get to a successful program. An assistant that makes long suggestions will occasionally be correct, however, more often than not the human will waste effort checking if an incorrect suggestion is correct. The DPR for a particular solution is defined as:

$$\text{DPR} = 1_{\text{Correct Solution}} \cdot \gamma^{\alpha \cdot \text{Tokens Read} + \beta \cdot \text{Tokens Written}} \tag{9}$$

The constant $\alpha$ specifies how "difficult" it is for the human to verify text that the assistant has suggested. Similarly, $\beta$ specifies how "difficult" it is for the human to write text on their own. Under this metric, the best assistant will help the human have the highest pass rate, while only suggesting completions which are most likely to be accepted and bring the program closest to its conclusion. This measures how useful the assistant is at generating *correct* solutions, not just how often it convinces the user to accept their flawed suggestion. To get the total DPR, we take the mean across all problems in the benchmark. In this work, we use $\gamma = 0.999$, $\alpha = 0.1$, and $\beta = 0.5$ to represent that it is often more difficult to generate than to verify, as well as to prevent the DPR of a long but correct solution from approaching 0.

Optimizing the DPR directly requires training with real human interaction data, or using an accurate human model, both of which are challenging. Our results show that empowerment is able to increase DPR and other metrics with an entirely offline dataset.

**Quantitative Results.** Comparisons between the baselines and our method with $\eta = 0.32$ are shown in Fig. 2, using Gemma-3-27B-it as the simulated human model. `Empower` outperforms all baselines on pass@1, accept ratio, and DPR. It is worth noting that the accept ratios of `Empower` and Base-10 are close for Llama-3.1-8B-Instruct and Qwen3-8B . We hypothesize that shorter suggestions are more likely to be accepted, which is why Base-10 has a higher acceptance rate. However, in that case, acceptance ratio does not correspond to a higher Pass@1, and therefore the DPR is lower. Just because a suggestion is short does not mean that it is correct. `Empower` tends to output suggestions which are more likely to be accepted and at the same time are also more likely to create correct programs.

We also perform the same set of experiments with Llama-3.3-70B-Instruct as the human model, for which we show results in Table 1 of Section B. `Empower` similarly beats the baseline on pass@1 and DPR. See Section B for the full numeric results.

### 5.3 Human Study: Evaluating Empowerment for Real-World Code Assistance

To evaluate empowerment at scale, we conducted an 18-person double-blinded user study in a code-generation setting with an assistant, similar to GitHub Copilot. Participants were randomly assigned to complete one of two python coding problems with corresponding testcases. The editor was configured to log whenever they accepted a suggestion or typed a character. This editor is very similar to Nano, and, unlike in the simulated results, allows the user to edit previous code. They first spent 25 minutes attempting the problem with no assistant. Then, they spent 15 minutes attempting the problem with Assistant 1, took notes on what they liked and didn't like about it, and then repeated

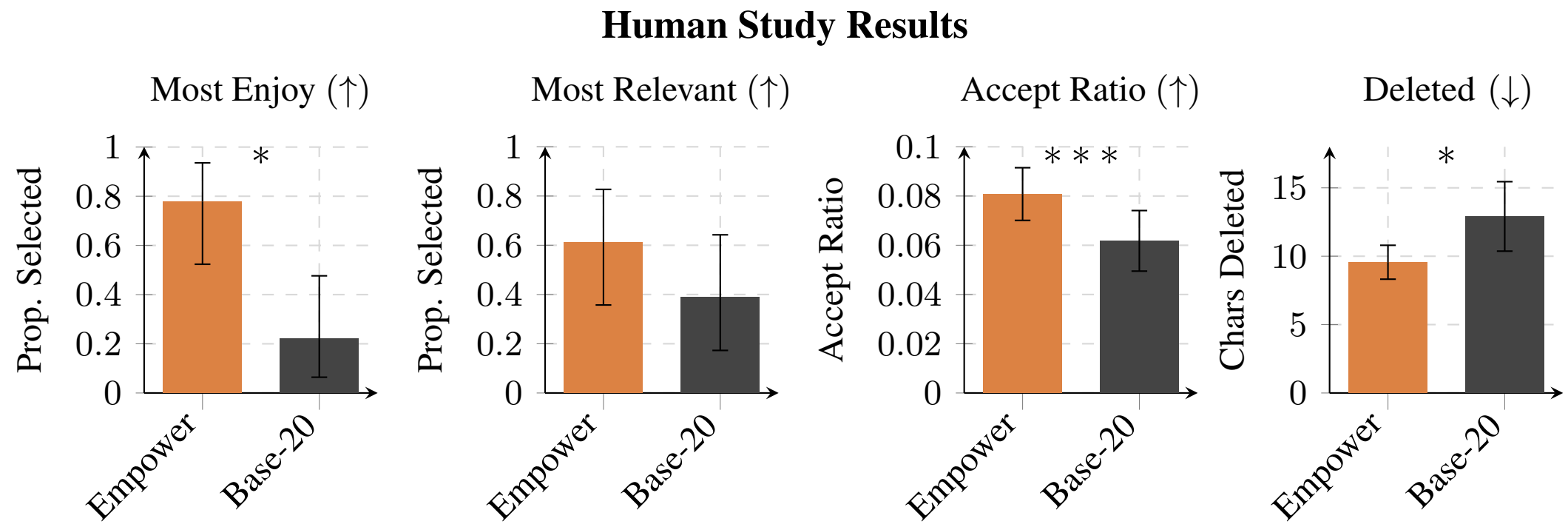

Figure 3: **Human study results with the Llama-3.1-8B-Instruct** assistant. Exact 95% confidence intervals are shown for Most Enjoy and Most Relevant as they represent Bernoulli data. Standard error bars are shown for Accept Ratio and Characters Deleted. In all cases, participants preferred using our `Empower` assistant.

this step for Assistant 2. Finally, they were asked to rank the assistants on several metrics including how relevant they found the suggestions and which assistant they would most enjoy using in practice. The order of the assistants was randomized and hidden from the researcher's view.

To choose which two assistants to compare, we ran a pilot study with Llama-3.1-8B-Instruct as the assistant. Participants in the pilot tended to prefer `Empower` with $\eta = 4$, and the Base-20 baseline, so we chose these to focus on for the full study.

**Survey Results.** We show the results of the study in Fig. 3. Participants ranked the `Empower` assistant as the one they would more enjoy using in practice 78% of the time, preferring `Empower` with a p-value of 0.015. Additionally, they ranked our assistant as providing more relevant suggestions 61% of the time, although the result was not statistically significant with a p-value of 0.240. Although both assistants tended to provide relevant suggestions, the `Empower` assistant was more judicious, providing fewer suggestions overall. Participants preferred this approach to assistance, which we attribute to the empowerment objective teaching the model to only complete as long as it is confident about what the user will type next.

**Quantitative Results.** We also collected quantitative data about the user-assistant interaction. The `Empower` assistant had an acceptance rate of 8.08% compared to the 6.18% of the Base-20 assistant. Participants accepted suggestions from our assistant more with $p = 0.0002$. Participants also tended to delete more accepted text from the Base-20 assistant than from ours. The average number of deleted characters per accepted suggestion was 12.91 for Base-20 and 9.56 for ours with $p = 0.0118$. On average, `Empower` suggested $\sim$208 suggestions per user, whereas Base-20 suggested $\sim$333. The baseline also tended to give longer suggestions, at 82.2 characters per suggestion compared to 43.6 for `Empower`. These differences highlight the type of assistance that empowerment enables. Rather than making decisions for the human, our empowerment objective trains an assistant that completes the obvious and no more. This leads to a more natural interaction, and reduces the feeling of frustration that comes from an assistant completing too much.

## 6 Discussion

In this paper, we showed how assistive (LLM) agents can provide their own feedback signal for learning by estimating how empowered a human coder is. Our logit threshold method tractably computes empowering suggestions, which maximize the impact that the human will have.

While we demonstrated success in coding assistance, we expect that LLM assistants trained with empowerment can be useful in many other domains, such as writing assistance or navigating an application. These also include more agentic applications where the assistant can infer when the human would predictably take an action, and instead take the action automatically. Our work enables the training of these agents at scale by simply configuring the likelihood estimator for a given domain.

While there has been much discussion of LLM post-training methods in recent years, there has been relatively less discussion of how these post-training methods are connected with the training objectives of the underlying LLMs. LLMs are trained primarily on next-token prediction, a self-supervised objective. Our work suggests that, in addition to training the base LLM with a self-supervised objective, the post-training (i.e., alignment) might also be done with a self-supervised objective.

**Limitations.** All experiments were conducted on competitive programming problems. Real-world code will often differs significantly in style and difficulty, which may require a more robust marginal likelihood estimator. The application of empowerment to more general coding tasks is left for future work.

As we have shown, our empowerment estimator works well for a coding environment where the human has full control over the future state. The estimator may become less accurate in an environment where some actions have more variable impact on the future than others.

## Reproducibility

To ensure that our results are reproducible, we provide a link to our code in Section A. The algorithm we used is described in Section 4.2, and the exact prompts we used for the assistants are detailed in Section D. The study instructions we provided to users, as well as the two problems they attempted, are given in Section E.

## Ethics Statement

The human study was conducted with Institutional Review Board (IRB) approval .

Empowerment methods may be used to create better assistive agents, improving the experience of people who collaborate with LLMs. There is a risk of an assistant being trained to self-empower, which would create a general power-seeking agent. However, our methods are focused on human-AI collaboration, which does not pose this risk.

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

## A Website and Code

The website that links to the code and configs to reproduce our experiments can be found at `https://anonymous.4open.science/r/codegen-384F/`.

## B Additional Simulated Results

Full experimental results are presented in Tables 1 and 2. We ablated the choice of human model, also training models on a Llama-3.3-70B-Instruct generated dataset and using it as the simulated human. We ablate the choice of $\eta$ and show results in Table 3. We train a Rejection Fine-Tuned (RFT) baseline on a single round of accepted suggestions to Codeforces problems, and include the results in Table 3 (Yuan et al., 2023). Table 3 also includes an evaluation of GPT-5-mini on our simulated environment.

## C Training Details

Experiments were performed on a NVIDIA H100 node with 8 GPUs, each with 80GB of VRAM. Pre-trained weights were taken from the LLaMA-3.1-8B, LLaMA-3.3-70B (Grattafiori et al., 2024), Qwen3-8B , Qwen3-14B (Yang et al., 2025), and Gemma-3-27B-it (Team et al., 2025) models, as described in Section 5. We finetuned the assistant for one epoch on a dataset of 4,138 examples with a test split size of 0.2.

The LlaMA models were used under the Llama 3.1 Community License Agreement. The Qwen models were used under the Apache 2.0 license. Gemma was used under the Gemma Terms of Use. Our training dataset was initialized from MatrixStudio/Codeforces-Python-Submissions, `https://huggingface.co/datasets/MatrixStudio/Codeforces-Python-Submissions`.

| Base Model | Name | Pass@1 (↑) | Accept Ratio (↑) | Discounted Pass Rate (↑) |
|---|---|---|---|---|
| Qwen3-8B | Empower | **0.218**$^{(\pm 0.019)}$ | 0.488$^{(\pm 0.024)}$ | **0.176**$^{(\pm 0.016)}$ |
| Qwen3-8B | SFT-20 | 0.167$^{(\pm 0.018)}$ | 0.192$^{(\pm 0.018)}$ | 0.116$^{(\pm 0.013)}$ |
| Qwen3-8B | SFT-10 | 0.152$^{(\pm 0.017)}$ | 0.299$^{(\pm 0.021)}$ | 0.114$^{(\pm 0.013)}$ |
| Qwen3-8B | SFT-RAND-1-30 | 0.156$^{(\pm 0.017)}$ | 0.201$^{(\pm 0.019)}$ | 0.109$^{(\pm 0.012)}$ |
| Qwen3-8B | Base-10 | 0.198$^{(\pm 0.019)}$ | **0.592**$^{(\pm 0.023)}$ | 0.162$^{(\pm 0.015)}$ |
| Qwen3-8B | Base | 0.183$^{(\pm 0.018)}$ | 0.351$^{(\pm 0.022)}$ | 0.143$^{(\pm 0.014)}$ |
| Llama3.1-8B Instruct | Empower | **0.282**$^{(\pm 0.021)}$ | 0.317$^{(\pm 0.022)}$ | **0.208**$^{(\pm 0.016)}$ |
| Llama3.1-8B Instruct | SFT-20 | 0.097$^{(\pm 0.014)}$ | 0.165$^{(\pm 0.017)}$ | 0.066$^{(\pm 0.010)}$ |
| Llama3.1-8B Instruct | SFT-10 | 0.104$^{(\pm 0.014)}$ | 0.257$^{(\pm 0.021)}$ | 0.074$^{(\pm 0.010)}$ |
| Llama3.1-8B Instruct | SFT-RAND-1-30 | 0.112$^{(\pm 0.015)}$ | 0.184$^{(\pm 0.018)}$ | 0.075$^{(\pm 0.010)}$ |
| Llama3.1-8B Instruct | Base-10 | 0.156$^{(\pm 0.017)}$ | **0.537**$^{(\pm 0.023)}$ | 0.127$^{(\pm 0.014)}$ |
| Llama3.1-8B Instruct | Base | 0.170$^{(\pm 0.018)}$ | 0.297$^{(\pm 0.021)}$ | 0.134$^{(\pm 0.014)}$ |
| Qwen3-14B | Empower | **0.249**$^{(\pm 0.020)}$ | 0.459$^{(\pm 0.023)}$ | **0.201**$^{(\pm 0.016)}$ |
| Qwen3-14B | SFT-20 | 0.145$^{(\pm 0.017)}$ | 0.188$^{(\pm 0.018)}$ | 0.102$^{(\pm 0.012)}$ |
| Qwen3-14B | SFT-10 | 0.165$^{(\pm 0.017)}$ | 0.292$^{(\pm 0.021)}$ | 0.126$^{(\pm 0.013)}$ |
| Qwen3-14B | SFT-RAND-1-30 | 0.145$^{(\pm 0.017)}$ | 0.226$^{(\pm 0.020)}$ | 0.106$^{(\pm 0.012)}$ |
| Qwen3-14B | Base-10 | 0.174$^{(\pm 0.018)}$ | **0.597**$^{(\pm 0.023)}$ | 0.143$^{(\pm 0.015)}$ |
| Qwen3-14B | Base | 0.161$^{(\pm 0.017)}$ | 0.299$^{(\pm 0.021)}$ | 0.127$^{(\pm 0.014)}$ |

Table 1: **Assistant results with Llama-3.3-70B-Instruct as the human model**. We evaluate on 554 LiveCodeBench problems, and find that `Empower` outperforms all baselines in terms of Pass@1 and DPR. Standard errors are shown in parentheses.

| Base Model | Name | Pass@1 (↑) | Accept Ratio (↑) | Discounted Pass Rate (↑) |
|---|---|---|---|---|
| Qwen3-8B | Empower | **0.178**$^{(\pm 0.018)}$ | **0.630**$^{(\pm 0.023)}$ | **0.156**$^{(\pm 0.016)}$ |
| Qwen3-8B | SFT-20 | 0.086$^{(\pm 0.013)}$ | 0.299$^{(\pm 0.021)}$ | 0.072$^{(\pm 0.011)}$ |
| Qwen3-8B | SFT-10 | 0.101$^{(\pm 0.014)}$ | 0.367$^{(\pm 0.023)}$ | 0.086$^{(\pm 0.012)}$ |
| Qwen3-8B | Base-10 | 0.090$^{(\pm 0.013)}$ | 0.582$^{(\pm 0.023)}$ | 0.080$^{(\pm 0.012)}$ |
| Qwen3-8B | Base | 0.092$^{(\pm 0.014)}$ | 0.400$^{(\pm 0.023)}$ | 0.083$^{(\pm 0.012)}$ |
| Llama3.1-8B Instruct | Empower | **0.176**$^{(\pm 0.018)}$ | **0.670**$^{(\pm 0.022)}$ | **0.150**$^{(\pm 0.015)}$ |
| Llama3.1-8B Instruct | SFT-20 | 0.070$^{(\pm 0.012)}$ | 0.231$^{(\pm 0.020)}$ | 0.057$^{(\pm 0.010)}$ |
| Llama3.1-8B Instruct | SFT-10 | 0.062$^{(\pm 0.011)}$ | 0.268$^{(\pm 0.021)}$ | 0.053$^{(\pm 0.010)}$ |
| Llama3.1-8B Instruct | Base-10 | 0.064$^{(\pm 0.011)}$ | 0.649$^{(\pm 0.022)}$ | 0.057$^{(\pm 0.010)}$ |
| Llama3.1-8B Instruct | Base | 0.064$^{(\pm 0.011)}$ | 0.383$^{(\pm 0.023)}$ | 0.055$^{(\pm 0.010)}$ |
| Qwen3-14B | Empower | **0.170**$^{(\pm 0.018)}$ | **0.659**$^{(\pm 0.022)}$ | **0.148**$^{(\pm 0.015)}$ |
| Qwen3-14B | SFT-20 | 0.088$^{(\pm 0.013)}$ | 0.381$^{(\pm 0.023)}$ | 0.077$^{(\pm 0.012)}$ |
| Qwen3-14B | SFT-10 | 0.101$^{(\pm 0.014)}$ | 0.461$^{(\pm 0.023)}$ | 0.088$^{(\pm 0.012)}$ |
| Qwen3-14B | Base-10 | 0.062$^{(\pm 0.011)}$ | 0.530$^{(\pm 0.023)}$ | 0.055$^{(\pm 0.010)}$ |
| Qwen3-14B | Base | 0.086$^{(\pm 0.013)}$ | 0.312$^{(\pm 0.022)}$ | 0.077$^{(\pm 0.012)}$ |

Table 2: **Assistant results with Gemma-3-27B-it as the human model.** We evaluate on 554 LiveCodeBench problems. We find `Empower` outperforms all baselines in terms of Pass@1 and DPR. Standard errors are shown in parentheses.

| Base Model | Name | Pass@1 (↑) | Accept Ratio (↑) | Discounted Pass Rate (↑) |
|---|---|---|---|---|
| GPT-5-mini | GPT-5-mini | **0.249**$^{(\pm 0.020)}$ | **0.699**$^{(\pm 0.021)}$ | **0.203**$^{(\pm 0.017)}$ |
| Llama3.1-8B Instruct | RFT | 0.057$^{(\pm 0.011)}$ | 0.384$^{(\pm 0.023)}$ | 0.049$^{(\pm 0.009)}$ |
| Llama3.1-8B Instruct | $\eta = 0.32$ | **0.176**$^{(\pm 0.018)}$ | **0.670**$^{(\pm 0.022)}$ | **0.150**$^{(\pm 0.015)}$ |
| Llama3.1-8B Instruct | $\eta = 0.5$ | 0.174$^{(\pm 0.018)}$ | 0.627$^{(\pm 0.023)}$ | 0.149$^{(\pm 0.015)}$ |
| Llama3.1-8B Instruct | $\eta = 1$ | 0.132$^{(\pm 0.016)}$ | 0.537$^{(\pm 0.023)}$ | 0.116$^{(\pm 0.014)}$ |
| Llama3.1-8B Instruct | $\eta = 2$ | 0.117$^{(\pm 0.015)}$ | 0.524$^{(\pm 0.023)}$ | 0.104$^{(\pm 0.014)}$ |
| Llama3.1-8B Instruct | $\eta = 4$ | 0.084$^{(\pm 0.013)}$ | 0.473$^{(\pm 0.023)}$ | 0.073$^{(\pm 0.011)}$ |
| Llama3.1-8B Instruct | SFT-20 | 0.070$^{(\pm 0.012)}$ | 0.231$^{(\pm 0.020)}$ | 0.057$^{(\pm 0.010)}$ |
| Llama3.1-8B Instruct | SFT-10 | 0.062$^{(\pm 0.011)}$ | 0.268$^{(\pm 0.021)}$ | 0.053$^{(\pm 0.010)}$ |
| Llama3.1-8B Instruct | Base-10 | 0.064$^{(\pm 0.011)}$ | 0.649$^{(\pm 0.022)}$ | 0.057$^{(\pm 0.010)}$ |
| Llama3.1-8B Instruct | Base | 0.064$^{(\pm 0.011)}$ | 0.383$^{(\pm 0.023)}$ | 0.055$^{(\pm 0.010)}$ |

Table 3: **Additional baselines and $\eta$ ablations with Gemma-3-27B-it as the human model.** We evaluate GPT-5-mini as an assistant. We also evaluate a Rejection Fine-Tuned (RFT) baseline. Our method with any choice of $\eta$ outperforms all baselines, except for GPT-5-mini, on Pass@1 and DPR. Base-10 has a higher acceptance rate than most choices of $\eta$, except for $\eta = 0.32$. Standard errors are shown in parentheses.

We de-duplicated the problems and re-generated the solutions using the corresponding human model that was being assisted.

## D Prompts

We provide prompts used for the LLMs in our experiments.

### D.1 Assistant Prompt

The assistant system prompt is:

```
You are assisting a human in a python code generation task. Your
    role is to provide suggested completions given
what they have already typed. Please try to infer what the human
    wants the next piece of code to be given the
code they have already written. If they have not written any code,
    please provide a good start to their program, such as with
    import statements or function definitions.

The way you will compose your suggestion is by providing the next
    version of the code which would replace the current code.
Please re-type the current code and then add in your suggested
    completion.
DO NOT output any other text, including no quotation marks.

## Remember to always re-type the code written so far and then add
    in your suggested completion.
If you don't re-type the code written so far *exactly as it is
    written* (with all of the functions, comments, import statements
    , etc)
an error will be raised.
```

The assistant user prompt is:

```
Now it's your turn! Please provide a completion for the following
    code:
```python
{{ code_to_complete }}
```
```

### D.2 Few Shot Examples for Assistant

```python
# > user
# > Now it's your turn! Please provide a completion for the following code:
# ```python
def twoSum(self, nums: List[int], target: int) -> List[int]:
    numMap = {}
    n = len(nums)

    # Build the hash table
    for i in range(n):
        numMap[nums[i]] = i

    # Find the complement
    for i in range(n):

# ```
# > assistant
# > Here is my suggested completion:
# ```python
def twoSum(self, nums: List[int], target: int) -> List[int]:
    numMap = {}
    n = len(nums)

    # Build the hash table
    for i in range(n):
        numMap[nums[i]] = i

    # Find the complement
    for i in range(n):
        complement = target - nums[i]
# ```
# > user
# > Now it's your turn! Please provide a completion for the following code:
# ```python
def whoami(name:
# ```
# > assistant
# > Here is my suggested completion:
# ```python
def whoami(name: str, age: int) -> str:
# ```
```

### D.3 Human Model Prompts

**Human Appender Prompt.** The prompts given to the human when they are deciding what to write next. They are provided both a system prompt and a user prompt. The system prompt is the following:

```
You are an expert Python programmer. You will be given a question (
    problem specification) and will generate a correct Python
    program that matches the specification and passes all tests.
*Please do not provide any sample outputs or testcases in your
    response. Additionally, you are only allowed to solve the
    problem *ONCE*.
Do not attempt to retry your solution if you are unhappy with it.
For example, if your solution is in a function called `solve`, you
    should only define one function called `solve`. DO NOT try to
    retry it if you think it has a bug.
For example, you should not write `solve2` if you think `solve` has
     a bug. Only the first solution will be counted, so simply stop
    writing once the first solution is finished--even if it is not
    correct.
```

The user prompt is the following:

```
### Question:
{{problem.question_content}}

### Format:
{
You will use the following starter code to write the solution to
    the problem and enclose your code within delimiters.
```python
{{ problem.starter_code }}
```
{
Read the inputs from stdin solve the problem and write the answer
    to stdout (do not directly test on the sample inputs). Enclose
    your code within delimiters as follows. Ensure that when the
    python program runs, it reads the inputs, runs the algorithm and
     writes output to STDOUT.
```python
# YOUR CODE HERE
```
{
```

**Human Acceptor Prompt.** When the human is choosing whether or not to accept or reject a suggestion, it has a different prompt. There is a system prompt, a user prompt that encourages it to reason about the suggestion, and finally a user prompt that asks it for its final decision to accept, reject, or finish. The system prompt is the following:

```
You are an expert Python programming agent. You will be given a
    question (problem specification) and will generate a correct
    Python program that matches the specification and passes all
    tests.

The way you will compose your program is by interacting with the
    file editing interface through tools.
In addition, you will be assisted by an automated assistant that
    will sometimes propose suggested completions to whatever you are
     typing.
At every step you will first write your thoughts on the suggestion
    and whether it correctly continues your solution, or instead
    introduces bugs.
Please *only consider the assistant suggestion* in your decision to
     accept the suggestion, reject the suggestion, or finish and
    submit your code for testing.
Previous code that you have written is not able to be changed, so
    you should only look at the changes the assistant is proposing.
At the very end of your thought process, you will write one word to
     specify which action you are taking.
Unless you take the "finish" action, you will be prompted
    afterwards to write what you would like to append to your
    program.
Therefore, please accept suggestions as long as they do not
    introduce bugs, and either help you solve the problem or improve
     the quality of your code.
It's ok if the the suggested completion is incomplete, because you
    can always append to it later.

You will first be prompted to write your thoughts on the suggestion
    .
Afterwords, you will be prompted to write which action you would
    like to take.

Here are the actions you have access to:
### "accept"
```

```
    description: Accepts a suggested completion given by an intelligent assistant. The suggested completion will then be incorporated into the code you have written.
### "reject"
    description: Rejects a suggested completion given by an intelligent assistant. The suggested completion will not be incorporated into the code you have written.
### "finish"
    description: Tells the editor that you have finished writing the program and to run the testcases. Only call this tool if you are confident that your program is finished. You will not be prompted to write any more code after calling this tool.

Remember, you will be able to continue writing your program regardless of whether you accept the suggested completion or not.
As long as the suggested completion does not introduce bugs, and either helps you solve the problem or improves the quality of your code, you should accept it.
DO NOT reject a suggestion because it is "minor" or "short". Only reject the suggestion if it is wrong, introduces a bug, or otherwise sets you back.
```

The user reasoning prompt is the following:

```
## You have written the following code:
```python
{code}
```

## Suggested Completion
Here is what your code would look like with a suggested completion:
```python
{suggestion}
```

## Suggested Completion diff
For clarity, here is the diff between your current code and the suggested completion code:
{git_diff_string(code, suggestion)}

## Instructions:
What do you think of the suggested completion? Do you think it is solving the question correctly, or does it introduce a bug or error?
Please write down your thoughts. You are not allowed to write any new code in your response, only your thoughts on whether the suggested completion helps you on your way to solving the problem, or otherwise improves the quality of your code.
It is ok if the the suggested completion is incomplete, because you will be prompted to append to it later.
You are also not able to take any actions at this stage.
```

After it has provided reasoning for whether or not it believes the suggestion is a good one, we prompt it to make its final decision with the following user prompt:

```
Now, please write which action you would like to take.
Remember, the actions available are "accept" to accept the suggested completion, "reject" to reject the suggested completion, and "finish" to finish writing your code and run the tests.
Please only call "finish" if you are confident that your code is correct and you are ready to run the tests.
```

# E Human Study

## E.1 Study Instructions

### Setup

1. Sign research consent form.
2. Run locally: `git clone redacted`
3. If on Mac:
   3.1. Navigate to the cloned repository and run: `./install.sh`
4. If on Windows:
   4.1. Install node: `https://nodejs.org/en/download`
   4.2. Run `npm install`
   4.3. Run `npm start`
5. If on Linux:
   5.1. Run `sudo apt install nodejs npm`
6. Enter your name in the box.
7. Switch the assistant to **Assistant 1**. Open up the scratchpad, type a few things, and make sure that a suggestion appears (suggestions will not always appear).
8. You can accept suggestions using the `Tab` key.
9. You can explicitly reject a suggestion using the `Esc` key.
10. Click **Back to Launch** at the top of the window.
11. Switch the assistant to **No Assistant**.
12. Move on to the Study section.

### Study

1. You are only allowed to use the Python docs: `https://docs.python.org/3/`. You may not use anything else on the internet.
2. To run the test cases:
   2.1. Save your file (`CMD + S`).
   2.2. At the top of the editor, click **Run Testcases**. This will copy a command to your clipboard which you can then paste and run in your terminal.
3. Set a timer for 25 minutes.
4. Switch the assistant to **No Assistant**.
5. Begin the problem.
6. Whenever you or the timer finish, switch to **Assistant 1**.
7. Open up the same problem.
8. Set a timer for 15 minutes and solve the problem with Assistant 1. Pay attention to what you like and dislike about this assistant.
9. Fill out your notes in this form: redacted.
10. Save the file you are working on (`CMD + S`).
11. Repeat steps 6–10 for Assistant 2.
12. Complete the rest of the form and rank the assistants.
13. Make sure to zip and upload your `problems` directory to the form.

## E.2 PROBLEM 1: LAVA TRAP

Simulate a single player walking on a square grid with lava squares. After each command, print if they fell into the lava, or, if they survived, print the player's current row, column, and facing.

The player will never move out of bounds of the grid. The top left of the grid is $(1, 1)$ and the bottom right is $(N, N)$.

### BOARD

- An $N \times N$ grid of characters:
  - `.` — empty cell
  - `L` — lava
- Cells are 1-indexed: row $1..N$, column $1..N$.

### PLAYER

- Starts at row $r$, column $c$, facing $dir \in \{\texttt{U}, \texttt{D}, \texttt{L}, \texttt{R}\}$.

### COMMANDS

You are given $Q$ commands:

1. `MOVE`
   Move forward one step in the current direction.
2. `FACE X` where $X \in \{\texttt{U}, \texttt{D}, \texttt{L}, \texttt{R}\}$
   Set the facing direction.

### TILE EFFECTS (AFTER THE MOVE)

- If the player moves into lava, the simulation ends, and `Game Over` is printed.

### INPUT

The input will come from standard in:

1. $N$
2. $N$ lines of grid (each of length $N$)
3. $r\ c\ dir$
4. $Q$
5. $Q$ lines of commands

### CONSTRAINTS

- $2 \leq N \leq 50$
- $1 \leq r, c \leq N$
- $dir \in \{\texttt{U}, \texttt{D}, \texttt{L}, \texttt{R}\}$
- Commands:
  - `MOVE`
  - `FACE U|D|L|R`
- $1 \leq Q \leq 2 \times 10^5$

OUTPUT

After each command, print one line:

```
r c dir
```

(with the player's 1-indexed row/col and facing as U|D|L|R).

EXAMPLE

**Input:**

```
3
..L
...
...
2 2 R
3
MOVE
FACE U
MOVE
```

**Output:**

```
2 3 R
2 3 U
Game Over
```

STARTER CODE

```python
import sys

def read_grid():
    """Reads N and then N lines of the grid. Returns (N, grid)."""
    n = int(sys.stdin.readline().strip())
    grid = [list(sys.stdin.readline().strip()) for _ in range(n)]
    return n, grid

def read_starting_position():
    """Reads r, c, dir. Returns (r, c, dir)."""
    parts = sys.stdin.readline().split()
    r, c, d = int(parts[0]), int(parts[1]), parts[2]
    return r, c, d

def read_q():
    """Reads q from stdin."""
    return int(sys.stdin.readline().strip())

def read_next_move():
    """Reads and returns the next command as a string, or None if EOF."""
    line = sys.stdin.readline()
    if not line:
        return None
    return line.strip()

def main():
```

### E.3 PROBLEM 2: SPECIAL KEYBOARD

Simulate a user typing on a special keyboard. They will type one character at a time. After they have finished typing, print what they wrote.

### INPUT

The input will come from standard in:

1. The number of characters that the user will type, $q$ ($1 \leq q \leq 2000$).
2. One character that the user types per line.
3. Characters may include letters, digits, spaces, punctuation, and the markers below.

### OUTPUT

- One line: the transformed string.

### SPECIAL TOGGLES

Most keyboards have a Caps Lock key that toggles between lowercase and uppercase letters. This special keyboard has that, in addition to several non-standard toggles. When the user types a special toggle key, turn the toggle **on**, and apply its rule for all of the text that the user types until they type the special toggle key again to turn it **off**.

- Toggle keys do not affect previously written text, only future text.
- Do not append the toggle character to the user's output.
- More than one toggle may be active at the same time.

### TOGGLE RULES

- `^` → Caps Lock: uppercase all letters while this toggle is active. (In Python: `s.upper()`)
- `~` → While active, consonants (letters that are not vowels) are duplicated, preserving case. ("y" counts as a consonant.)
- `#` → While active, only digits and the *first* "`.`" encountered are appended to the output.
  - Skip all other characters.
  - If a second "`.`" appears (or any additional one), skip it.

  (In Python: check if a character is a digit with `s.isdigit()`.)

### EXAMPLES

**Input**

```
5
^
a
b
^
c
```

**Output**

```
ABc
```

**Input**

```
9
^
a
~
b
c
```

```
~
d
^
e
```

**Output**

```
ABBCCDe
```

**Input**

```
18
^
d
~
b
#
I
6
7
.
9
.
1
Z
#
a
^
c
.
```

**Output**

```
DBB67.91Acc.
```

STARTER CODE

```python
import sys
from typing import List, Tuple

def read_q() -> int:
    """Read the number of typed characters (q) from the first line."""
    line = sys.stdin.readline()
    if not line:
        raise EOFError("Expected an integer q on the first line.")
    return int(line.strip())

def read_next_char() -> str:
    """
    Read the next 'character per line'.
    """
    line = sys.stdin.readline()
    if line == "":
        raise EOFError("Unexpected end of input while reading characters.")
    # Take the first character on the line.
    return line[0]

def main() -> None:
```

### E.4 Questionnaire

This questionnaire was given to participants through a Google Form.

1. What is your name?
2. Which question are you solving?
3. Assistant 1 Notes (Paragraph entry).
4. Assistant 2 Notes (Paragraph entry).
5. How relevant are the assistant's suggestions? (Assign each label to only one assistant)
   (a) Assistant 1. [1 (Most relevant suggestions) or 2 (Least relevant suggestions)]
   (b) Assistant 2. [1 (Most relevant suggestions) or 2 (Least relevant suggestions)]
6. How often did you have to delete the assistant's work? (Assign each label to only one assistant)
   (a) Assistant 1. [1 (Fewest deletes) or 2 (Most deletes)]
   (b) Assistant 2. [1 (Fewest deletes) or 2 (Most deletes)]
7. Which would you most enjoy using in practice? (Assign each label to only one assistant)
   (a) Assistant 1. [1 (Most enjoy) or 2 (Least enjoy)]
   (b) Assistant 2. [1 (Most enjoy) or 2 (Least enjoy)]

### E.5 Additional Results

In the survey we asked participants to rank the assistants based on how often they had to delete the assistant's work. In total, 17 out of the 18 participants ranked our `Empower` method over the baseline ($p = 0.00007$). As we also collected the participant's keypresses, we instead included the exact number of characters which were accepted and later deleted in the main text, which is a more informative metric.

### E.6 Example Final Programs

We include examples of final programs written by two randomly chosen human study participants. The code is highlighted to indicate the source of each character: the starter code is colored beige, the participant code is colored blue, and the assistant code is colored green.

#### E.6.1 Lava Trap Base-20

The final program written by a participant with the Base-20 assistant on the Lava Trap problem.

USER ASSISTANT STARTER_CODE

```
import sys

def read_grid():
    """Reads N and then N lines of the grid. Returns (N, grid)."""
    n = int(sys.stdin.readline().strip())
    grid = [list(sys.stdin.readline().strip()) for _ in range(n)]
    return n, grid

def read_starting_position():
    """Reads r, c, dir. Returns (r, c, dir)."""
    parts = sys.stdin.readline().split()
    r, c, d = int(parts[0]), int(parts[1]), parts[2]
    return r, c, d

def read_q():
    """Reads q from stdin."""
    return int(sys.stdin.readline().strip())
```

```
def read_next_move():
    """Reads and returns the next command as a string, or None if EOF."""
    line = sys.stdin.readline()
    if not line:
        return None
    return line.strip()

def main():
    q = read_q()
    n, grid = read_grid()
    r,c,d = read_starting_position()
    for _ in range(q):
        print(f"this is q{q}")
        next_move = read_next_move()
        print("running this here")
        # Process the move
        # For example, let's assume we want to print the positi
        # Process the move here, for example:
        if next_move == "MOVE":
            # Get the direction
            print("running this")
            # Move in the given direction
            if d == "U":
                r -= 1
            elif d == "D":
                r += 1
            elif d == "L":
                c -= 1
            elif d == "R":
                c += 1
            if grid[r-1][c-1] == "L":
                print("Game Over")
                return
            else:
                print(f"{r} {c} {d}")
        else:
            d = next_move[-1]
            print(f"{r} {c} {d}")
```

### E.6.2 Lava Trap Empower

The final program written by the same participant with the Empower assistant on the Lava Trap problem.

USER ASSISTANT STARTER_CODE

```
import sys

def read_grid():
    """Reads N and then N lines of the grid. Returns (N, grid)."""
    n = int(sys.stdin.readline().strip())
    grid = [list(sys.stdin.readline().strip()) for _ in range(n)]
    return n, grid

def read_starting_position():
    """Reads r, c, dir. Returns (r, c, dir)."""
    parts = sys.stdin.readline().split()
    r, c, d = int(parts[0]), int(parts[1]), parts[2]
    return r, c, d

def read_q():
```

```
    """Reads q from stdin."""
    return int(sys.stdin.readline().strip())

def read_next_move():
    """Reads and returns the next command as a string, or None if EOF."""
    line = sys.stdin.readline()
    if not line:
        return None
    return line.strip()

def main():
    n, grid = read_grid()
    r, c, dir = read_starting_position()
    q= read_q()

    while q >=0:
        next_dirs = {"L":-1, "R": 1, "U": -1, "D": 1}
        m = read_next_move()
        if not m:
            return
        if m == "MOVE":
            if dir in ["U", "D"]:
                r += next_dirs[dir]
            else:
                c += next_dirs[dir]
            if grid[r-1][c-1] == "L":
                print("Game Over")
                return
            else:
                print(f"{r} {c} {dir}")
        else:
            new_dir = m[-1]
            dir = new_dir
            print(f"{r} {c} {dir}")
        q -= 1

main()
```

### E.6.3 Special Keyboard Base-20

The final program written by a different participant with the Base-20 assistant on the Special Keyboard problem.

USER | ASSISTANT | STARTER_CODE

```
import sys
from typing import List, Tuple

def read_q() -> int:
    """Read the number of typed characters (q) from the first line."""
    line = sys.stdin.readline()
    if not line:
        raise EOFError("Expected an integer q on the first line.")
    return int(line.strip())

def read_next_char() -> str:
    """
    Read the next 'character per line'.
    """
    line = sys.stdin.readline()
```

```python
    if line == "":
        raise EOFError("Unexpected end of input while reading characters.")
    # Take the first character on the line.
    return line[0]

def main() -> None:

    output = ''

    # Read the number of characters
    q = read_q()

    caps_flag = False
    duplicate_flag = False
    digits_flag = False
    saw_dot = False

    # Read and process
    for _ in range(q):
        char = read_next_char()

        if char == '^':
            caps_flag = not caps_flag
            continue
        if char == '#':
            digits_flag = not digits_flag
            saw_dot = False
            continue
        if char == '~':
            duplicate_flag = not duplicate_flag
            continue

        # Check if we need to process the character based on the flags
        if caps_flag:
            char = char.upper()
        if duplicate_flag:
            if char not in {'a', 'e', 'i', 'o', 'u'}:
                char = 2*char
        if digits_flag:
            if char == '.' and not saw_dot:
                saw_dot = True
            elif not char.isdigit():
                char = ''

        output += char

    print(output)

main()
```

### E.6.4 Special Keyboard Empower

The final program written by the different participant with the `Empower` assistant on the Special Keyboard problem.

USER ASSISTANT STARTER_CODE

```python
import sys
from typing import List, Tuple

def read_q() -> int:
```

```python
    """Read the number of typed characters (q) from the first line."""
    line = sys.stdin.readline()
    if not line:
        raise EOFError("Expected an integer q on the first line.")
    return int(line.strip())

def read_next_char() -> str:
    """
    Read the next 'character per line'.
    """
    line = sys.stdin.readline()
    if line == "":
        raise EOFError("Unexpected end of input while reading characters.")
    # Take the first character on the line.
    return line[0]

def main():

    output = ''
    caps_flag = False
    digits_flag = False
    duplicate_flag = False
    saw_dot = False

    q = read_q()

    for _ in range(q):
        char = read_next_char()

        if char == '^': caps_flag = 1 - caps_flag
        elif char == '~': duplicate_flag = 1 - duplicate_flag
        elif char == '#': digits_flag = 1 - digits_flag

        if digits_flag:
            if char == '.' and saw_dot = True:
                char = ''
            if char == '.' and saw_dot = False:
                saw_dot = True

            elif not char.isdigit():
                char = ''

        elif caps_flag:
            char = char.upper()
        elif duplicate_flag:
            if char not in {'a', 'e', 'i', 'o', 'u'}:
                char = char*2

        output += char

    return output

print(main())
```

## F LLM Acknowledgment

We did not use LLMs significantly in the writing or ideation of this paper. An LLM was used to proof-read, and a few sentences were reworded accordingly.