# OpenReview forum: "Training LLM Agents to Empower Humans"
_ICLR.cc/2026/Conference — Submitted to ICLR 2026_

### Official Review · Reviewer_RDVP · 2025-10-30

**Soundness:** 2
**Presentation:** 3
**Contribution:** 2
**Rating:** 4
**Confidence:** 3

**Summary:**

The paper proposes an LLM decoding method based on [Learning to Assist Humans without Inferring Rewards](https://arxiv.org/pdf/2411.02623), and evaluate it in a synthetic and a user study.

**Strengths:**

The paper proposes a novel decoding method based on likelihood, and does empirical evaluation of it. The application domain of coding is important.

**Weaknesses:**

I have concerns with the proposed method and empirical evaluation.
1. The main novelty methodologically over [Learning to Assist Humans without Inferring Rewards](https://arxiv.org/pdf/2411.02623) is their approximation of the mutual information. To me it is quite a coarse approximation to upper bound mutual information by entropy of one of the variables. Fundamentally, empowerment is about some causal influence, not entropy alone. Without the derivation this method looks like a likelihood-based stopping criterion, which does not resemble empowerment.
2. In the simulated human study, the human model is a strictly stronger (in terms of parameters) and the performance gains could just come from the LLM acting as the assistant "staying more passive", and letting the human model do the work. For empowerment, a main question is how the cognitive burden between the model and the human is allocated. If the human does most of the work, then this might lead to better performance. This makes it hard for me to evaluate the evidence.
3. I commend the authors on running a user study, but the main concern from the simulated study (does the model just make the human solve the task) is not addressed: The coding problems that users are shown are quite short, and it might be that the decoding method proposed here leads to humans essentially solving the task.
4. As the method relies on likelihood estimates of the LLM, calibration becomes an issue, which the paper does not adress.

**Questions:**

- How calibrated are uncertainty estimates of the model?
 - How does your decoding change outputs in terms of length? (Outputs should get shorter if I am getting it correctly, as the end-of-string token will still be output)
 - Can you rule out that the performance gains in the simulated human study comes from the assistant LLM "staying out of the game" / being more passive?
Typo: AH = {{ACCEPT}×L≤KH, {REJECT}×L≤KH, FINISH} doesn't type-check.

---

> ### Author Response · Authors · 2025-11-21
> **Author Response**
>
> Dear Reviewer,
>
> We thank the reviewer for the thoughtful feedback. The main concerns seem to be on the theoretical connection between our method and previous works, the possibility that the assistant is staying passive, and the calibration of the likelihood estimates. We have added a paragraph to Section 4.3 that discusses the theoretical limitations of our approximate bound and why we believe it is reasonable in our setting. We have also added several randomly selected samples of final programs from the user study to Appendix E.6, highlighting the contributions by the assistant and the user. We have also added an ablation over the $\eta$ parameter in [Table 3](http://res.cloudinary.com/dp7qzzmt2/image/upload/v1763875401/pyyutkjn9onwp2njxut8.png) that demonstrates good results on a range of choices. **Do these clarifications and results address the concerns?** We look forward to continuing the discussion.
>
> > Coarse approximation of mutual information
>
> This is true, it is a coarser approximation. However, the benefit of this approach is that we can use entirely offline data. Learning to Assist Humans without Inferring Rewards relies on online training and therefore a strong model of the human behavior. If this human model degrades their mutual information estimator will not transfer well to real humans. By contrast, our approximation suffers no such limitation. We evaluated the contrastive classifier used in that paper in our tab setting but found that it was unable to outperform the untrained models on an earlier version of LiveCodeBench. We attribute this to the fragility of learning an exact approximation for mutual information: it is difficult for the estimator to generalize.
>
> > Performance gains could come from the assistant ''staying passive''
>
> That's a great and very important question. This is why we introduced the Discounted Pass Rate (DPR) metric discussed in Section 5.2, to exactly capture how much the assistant is helping offload the human cognitive burden of reading and writing text. We found that our method has a much higher DPR than the baselines. We also have added several samples of final programs from the user study to Appendix E.6. These show that the assistants are making nontrivial contributions to the program, so the increase on all user study metrics is caused by our assistant being stronger, not by it staying passive.
>
> > How calibrated are uncertainty estimates of the model?
>
> Calibration was certainly a concern. Prior work has found that in some settings LLMs are good estimators of human variability, while in others they may overestimate the variability in a task ([Giulianelli et al., 2023](https://aclanthology.org/2023.emnlp-main.887/)). Other work has introduced methods to improve the calibration of an LLM ([Xie et al., 2024](http://arxiv.org/abs/2412.12767)). This is why in the simulated human evaluations we include several different assistant models, each calibrated to their own training process, working with two different humans. In all cases, our assistants were stronger than the baselines. This, combined with the results from the user study, provides evidence that our method is not very sensitive to the variation in calibration between models. Future work might investigate if additional calibration techniques improve the quality of the trained assistant.
>
> > How does your decoding change in terms of length?
>
> The outputs of a model trained with our method are shorter than the standard Instruct model. This is why we also evaluate with capped baselines, such as Base-10 and Base-20 which can only output up to 10 and 20 tokens respectively. Users in our pilot study preferred Base-20 over SFT-10 and the uncapped model, so we compared that with our method. In the full user study we found that our method tended to output suggestions that were 43.6 characters (not tokens) long, whereas Base-20 output 82.2 characters on average.
>
> > Typo
>
> Thank you for catching this, we have updated the PDF to clarify the action space construction.
>
> ---
>
> Giulianelli, M. et al., 2023. ''[What Comes Next? Evaluating Uncertainty in Neural Text Generators Against Human Production Variability](https://aclanthology.org/2023.emnlp-main.887/).'' *EMNLP*
>
> Xie, L. et al., 2024. ''[A Survey of Calibration Process for Black-Box LLMs](http://arxiv.org/abs/2412.12767).'' arXiv:2412.12767

---

### Official Review · Reviewer_Uf2z · 2025-11-01

**Soundness:** 3
**Presentation:** 3
**Contribution:** 3
**Rating:** 6
**Confidence:** 3

**Summary:**

This paper introduces Empower, a new unsupervised fine-tuning method for training language model assistants that aim to empower human users rather than replace them. Instead of optimizing for task completion or explicit feedback, the model learns to maximize the human’s empowerment—the ability of a user’s actions to influence future outcomes. Using a simple logit-threshold algorithm, the assistant learns to generate “obvious” completions (e.g., boilerplate code) while leaving key decisions to the user. In code generation experiments with LiveCodeBench, Empower significantly improves human–model collaboration, achieving higher Pass@1, acceptance rate, and a 192% improvement in simulated human success over supervised fine-tuning baselines. In a 18-participant user study, participants preferred Empower 78% of the time, citing fewer, more useful suggestions.

**Strengths:**

This paper offers a thoughtful and well-motivated direction for designing more cooperative AI assistants. The central idea—training models to maximize human empowerment rather than simply task completion—is timely. It reframes alignment from “doing the task” to “helping humans do the task better,” which feels like a meaningful step toward more human-centered AI.

The Empower algorithm itself is simple and scalable, relying only on offline data and language-model likelihoods, which makes it practical compared to methods requiring human feedback or reinforcement learning. Empirically, the results are solid: the method improves both automated coding benchmarks and human–AI collaboration metrics, and the inclusion of an 18-person user study adds credibility beyond simulation.

**Weaknesses:**

While the paper is well written and empirically convincing, its novelty is somewhat incremental from prior empowerment work in reinforcement learning and assistive robotics. The idea of empowerment—training an agent to maximize the user’s ability to influence the environment—is not new. It has already been explored for many years in reinforcement learning (RL) and human–robot collaboration research. The adaptation to LLMs mainly involves reinterpreting token likelihoods as proxies for empowerment, which may not fully capture the theoretical intent of empowerment-based control.

Methodologically, the logit-threshold rule feels a bit ad hoc. The paper doesn’t fully explain why this particular thresholding approach should reflect empowerment, or how sensitive the results are to the exact cutoff. It’s an intuitive idea—use token likelihoods to decide where the model should stop generating—but the theoretical link between “high likelihood” and “empowering the human” isn’t very clear. The method works in practice, but it feels more heuristic than principled.

The evaluation, while extensive in the coding domain, is narrowly scoped: all experiments focus on short-horizon code completion, and it's unclear how well the method generalizes to other real-world assistive settings (e.g., writing, planning, dialogue). The simulated “human” setup also relies on large LLMs as stand-ins for humans, which may inflate results.

**Questions:**

How sensitive are the results to the choice of the logit threshold (η)?

The current setup assumes the base LLM’s likelihood estimates are good proxies for human predictability—does this hold for weaker or domain-shifted models?

---

> ### Author Response · Authors · 2025-11-21
> **Author Response**
>
> Dear Reviewer,
>
> We thank the reviewer for the thoughtful feedback. The main concerns seem to relate to the theoretical connection between our method and previous works on empowerment, as well as the sensitivity to the choice of threshold. We have added a paragraph to Section 4.3 that discusses the theoretical limitations of our approximate bound and why we believe it is reasonable in our setting, as well as an ablation over the $\eta$ parameter in [Table 3](http://res.cloudinary.com/dp7qzzmt2/image/upload/v1763875401/pyyutkjn9onwp2njxut8.png) that demonstrates good results on a range of choices (changes highlighted in orange). **Do these clarifications and results address the concerns?** We look forward to continuing the discussion.
>
> > may not fully capture the theoretical intent of empowerment-based control
>
> We recognize that our method is only an approximation, and have added a paragraph to Section 4.3 that discusses these points. You are correct that there are elements of empowerment that we are not capturing with our approximation. Some previous applications of empowerment have been in environments where the agent's control over the environment varies depending on their state. Our method does not approximate this, because it is instead focused on the action likelihoods. For many text-based domains such as ours, the user has full control over the future state. In these settings, empowerment can naturally be approximated by predictability as we now show in Section 4.3.
>
> > The evaluation is narrowly scoped
>
> Although our evaluation is only focused on the coding domain, we believe that it is the best environment for testing this type of assistance, and serves as the canonical testbed in several well-received recent papers such as LiveCodeBench ([Jain et al., 2024](http://arxiv.org/abs/2403.07974)), ([Jimenez et al., 2024](http://arxiv.org/abs/2310.06770)), and ([Austin et al., 2021](http://arxiv.org/abs/2108.07732)). This is because we can easily verify the correctness of a program, not just whether a suggestion is accepted or rejected.
>
> > How sensitive are the results to the choice of the logit threshold ($\\eta$)?
>
> We have added a plot showing our ablation of $\\eta$ in [Table 3](http://res.cloudinary.com/dp7qzzmt2/image/upload/v1763875401/pyyutkjn9onwp2njxut8.png) in the appendix. Our method with $\\eta$ values of $0.32, 0.5, 1, 2,$ and $4$ outperforms all of the baselines on Pass@1 and Discounted Pass Rate (DPR). Base-10 has a higher acceptance rate than most choices of $\\eta$, except for $\\eta=0.32$. This aligns with our claim that our method improves assistance in a meaningful way that goes beyond increasing acceptance rate. Also, while our choice of $\\eta=0.32$ for the simulated environment may look over-optimized, it is actually the first $\\eta$ we tried, as it corresponds to a cumulative likelihood of 80%.
>
> > does this hold for weaker or domain-shifted models?
>
> It is possible to construct a domain where the LLM logits are not good proxies for the marginal human entropy. However, for most realistic text domains we hypothesize that LLMs will be reasonably calibrated due to their training on massive text datasets. In our evaluations with a simulated human, shown in Figure 2 and Table 1, we include evaluations with the stronger Qwen3-14B as well as the weaker Llama-3.1-8B-Instruct and find that our method performs well on both sizes.
>
> ---
>
> Austin, J. et al., 2021. ''[Program Synthesis With Large Language Models](http://arxiv.org/abs/2108.07732).'' arXiv:2108.07732
>
> Jain, N. et al., 2024. ''[LiveCodeBench: Holistic and Contamination Free Evaluation of Large Language Models for Code](http://arxiv.org/abs/2403.07974).'' arXiv:2403.07974
>
> Jimenez, C. E. et al., 2024. ''[SWE-Bench: Can Language Models Resolve Real-World GitHub Issues?](http://arxiv.org/abs/2310.06770).'' arXiv:2310.06770

---

### Official Review · Reviewer_DgmQ · 2025-11-01

**Soundness:** 3
**Presentation:** 2
**Contribution:** 2
**Rating:** 4
**Confidence:** 3

**Summary:**

This paper proposes a new approach to training assistive large language models (LLMs) by maximizing human empowerment. The model is trained to help users reach states where they have greater control and more meaningful choices.
The authors formalize empowerment through mutual information and introduce Empower to select completions whose cumulative likelihood exceeds a threshold.
The proposed approach is empirically validated in both simulated environments (LiveCodeBench) and an 18-participant human user study, demonstrating consistent improvements in performance and user preference.

**Strengths:**

- Considering LLMs as assistants is an interesting framing. Introducing empowerment as an alignment signal is an interesting and novel idea.
- The method relies only on offline text data, providing a potentially scalable and cost-effective alternative to RLHF or preference-based fine-tuning pipelines.

**Weaknesses:**

- The environment setup supports only linear text continuation: neither the simulated nor the human users can edit previously written code. This does not reflect how developers actually write programs, where they sometimes design high-level structures first, fill in details later, and revise earlier sections when bugs are found.
- For the user study, there is no performance comparison with RLHF-trained assistants. Without such baselines, it’s unclear how large the performance gap is and whether the proposed cost savings justify potential losses in performance.

**Questions:**

- Many existing products (e.g., GitHub Copilot) already perform line-by-line completion with real human users. Have you compared your Empower assistant against such commercial systems, or considered including them in the user study?
- How is the empowerment threshold chosen? Have you analyzed sensitivity to this hyperparameter?

---

> ### Author Response · Authors · 2025-11-21
> **Author Response**
>
> Dear Reviewer,
>
> Thank you for the thoughtful feedback. The main concerns seem to relate to our experimental setup and additional ablations. We would like to clarify that **our setup does allow editing past code**. We have also **added an experiment** comparing against GPT-5-mini and **added an ablation** to justify the selection of the $\eta$ parameter to [Table 3](http://res.cloudinary.com/dp7qzzmt2/image/upload/v1763792979/Codegen\_Rebuttal-1763792961905\_owkuij.png) in the Appendix. **Do these clarifications and results address your concerns?** We look forward to continuing the discussion.
>
> > neither the simulated nor the human users can edit previously written code
>
> We would like to clarify that our editor **does let the user go back and edit past code.** The only limitation to editing previous code is in the simulated environment, not in the user study. The user study editor is very similar to Nano. Please let us know if this is what your concern was referring to.
>
> > there is no performance comparison with RLHF-trained assistants
>
> While the focus of our work is on training assistants without human interaction or feedback, we have evaluated a simple rejection fine-tuned (RFT) Llama-3.1-8B-Instruct with the simulated gemma-3-27b-it human ([Yuan et al., 2023](http://arxiv.org/abs/2308.01825)). This model was finetuned only on suggestions that were accepted. It has a high acceptance ratio of 38%, although this is still lower than our method's acceptance ratio of 67%. It significantly underperforms in terms of Pass@1, sitting at 6% compared to our 18%, and DPR, at 0.049 compared to our 0.150. We have added these results to Table 3 in the Appendix.
>
> > compare against copilot
>
> Unfortunately, GitHub CoPilot and other commercial tab-completion assistants do not provide public APIs. Instead, we have evaluated GPT 5-Mini as an assistant in our simulated environment, using gemma-3-27b-it as the human model. GPT 5-Mini achieves a Pass@1 of 24.9%, an acceptance rate of 70%, and a DPR of 0.2031 +- 0.0167. This is better than any assistant performs with the gemma-3-27b-it human model, with our assistant coming in second. We have added these results to [Table 3](http://res.cloudinary.com/dp7qzzmt2/image/upload/v1763792979/Codegen\_Rebuttal-1763792961905\_owkuij.png) in the Appendix.
>
> > selection of $\\eta$ parameter
>
> We have added a plot showing our ablation of $\\eta$ in [Table 3](http://res.cloudinary.com/dp7qzzmt2/image/upload/v1763792979/Codegen\_Rebuttal-1763792961905\_owkuij.png) in the Appendix. Our method with any choice of $\\eta$ outperforms all of the baselines on Pass@1 and Discounted Pass Rate (DPR). Base-10 has a higher acceptance rate than most choices of $\\eta$, except for $\\eta=0.32$. This aligns with our claim that our method improves assistance in a meaningful way that goes beyond increasing acceptance rate. Also, while our choice of $\\eta=0.32$ for the simulated environment may look over-optimized, it is actually the first $\\eta$ we tried, as it corresponds to a cumulative likelihood of 80%.
>
> ---
>
> Yuan, Z. et al., 2023. ''[Scaling Relationship on Learning Mathematical Reasoning With Large Language Models](http://arxiv.org/abs/2308.01825).'' arXiv:2308.01825

---

### Author Response · Authors · 2025-12-02
**Summary of Reviews and Responses**

Dear AC,

We thank you for taking the time to review our work. Reviewers liked our novel application of empowerment to LLM alignment as well as the simplicity and offline nature of our method. They also appreciated the scale of our work, and our inclusion of a human study. Reviewer Uf2z highlighted that the empirical “results are solid: the method improves both automated coding benchmarks and human–AI collaboration metrics, and the inclusion of an 18-person user study adds credibility beyond simulation.” The main reviewer concerns seem to be the theoretical connection between our method and previous works on empowerment, the sensitivity of our results to our choice of $\eta$, a hyperparameter we introduced that influences the length completion that we train on, the possibility that our assistant simply learned to stay passive, and the lack of comparison to RLHF or commercial baselines.

**Theoretical Connection.** We have added a paragraph to Section 4.3 that makes the theoretical connection more explicit. Admittedly, our analysis is using the fact that in many text domains such as ours the user has full “control” over the state, because they alone choose how the text is modified. This is core to our method because it allows us to simplify the empowerment objective and make it tractable to optimize for. However, it is still only an approximation, and we have added text to make that more clear.

**Choice of $\eta$.** We have added an ablation over the choice of $\eta$ in Table 3 in the Appendix. It is important to note that our method outperforms the baselines in Pass@1 and Discounted Pass Rate–a measure of the quality of assistance–at all choices of $\eta$ we tried. The specific $\eta$ we use in the paper was not optimized for, and was the first one we tried. It corresponds to a cumulative likelihood threshold of 80%.

**Staying Passive.** In Appendix E.6, we have added transcripts from the user study with the text contributed by the user and assistant highlighted. These show that our assistant is not staying passive, and so our improvements are not attributable to the human doing more work. We would also like to highlight that for the simulated results we have introduced a metric, the Discounted Pass Rate, which is exactly intended to measure how much “work” the simulated human has to do to get a working program. It takes into account the amount of text that the simulated human writes themselves, as well as the amount of suggested text they have to read. Our method significantly outperforms the baselines on these fronts.

**RLHF Comparison.** We have added a comparison with a Rejection Fine Tuned (Yuan et al. 2023) baseline that only trains on accepted suggestions as well as with GPT-5-mini. These results are in Table 3. Our method outperforms the RFT baseline, and underperforms GPT-5-mini.

**Important Clarification.** Finally, reviewer DgmQ’s main concern seemed to be that our setup in the user study does not allow editing of past code. We have clarified that this is a misunderstanding, and that the participants collaborated with the assistant in a nano-style editor where they could modify old code. We have updated Section 5.3 to make this distinction clear.

---

### Meta-Review · Area_Chair_jv4D · 2026-01-03

**Summary:**

The paper proposes a method for training code completion models using offline human-written code data. The method is very simple: for each prefix, a completion is selected by taking the next $i$ tokens until the cumulative probability of the completion according to a language model is below a threshold. The idea is that this way we can identify obvious pieces of code that do not require human decisions, while more uncertain pieces are left to the user. The authors motivate this method through _empowerment_, which is an objective defined in information theoretic terms. Finally, the authors train assistant models using this objective, and show improved performance on automated evaluations (where an LLM serves as a user), and in a human study, compared to several custom baselines.

One of the big concerns of the reviewers (RDVP, Uf2z) was the limited connection between the theoretical empowerment formulation and the actual method proposed by the authors. I would also argue that the current presentation makes the method appear more complicated than it should be. The method is very simple: we just decide on the completion length so that the completion is within a confidence threshold according to a language model. It is intuitive that this procedure would select completions where little uncertainty is present, even without appealing to information theory, or the MDP formulation.

Arguably, confidence thresholding is something that is already used in production code completion systems. E.g., see [this blogpost](https://research.google/blog/ml-enhanced-code-completion-improves-developer-productivity/)
> For multi-line suggestions, we iteratively apply the single-line model with learned thresholds for deciding whether to start predicting completions for the following line.
Arguably, this thresholding is different: here the thresholding is applied at test-time to decide how many lines of the completion to show to the user; the authors do the thresholding at training time to generate the data for training the completion model.

Another concern raised by the reviewers (DgmQ) is the quality of the baselines and the evaluation setup. In particular, in the automated evaluation the code is generated sequentially, with no ability to update code that has already been generated. Importantly, this is not an issue in the human study, where the human participants were able to edit the code arbitrarily, so this concern is partially addressed.

Another issue with the evaluation is that the authors do not compare to any established methods for training code completion models, including RLHF. Production code-completion systems already include multiple tricks: in particular there are techniques for deciding when to show a completion proposal, and how long it should be (see above on the thresholding). The paper currently only compares to fairly naive baselines. This is partially justified by the fact that the authors are using a restrictive setting with no explicit human feedback, and it is unfair to expect them to outperform production-level systems. However, the weak baselines make it hard to assess whether the proposed method would be of practical significance.

In response to this concern, the authors compared there method to an off-the-shelf GPT 5-mini during the rebuttal, and found that GPT-5 mini performed better. It is not clear how this result should be interpret, as GPT-5-mini is likely a stronger model than the models finetuned by the authors, but also it did not undergo any specific code completion training.

**Reviewer Concerns:**

I described two concerns above: connection between method and empowerment motivation and quality of evaluation. I believe these concerns are not fully addressed by the rebuttal. The authors added additional material on the connection between empowerment and the method, and also clarified some of the issues with evaluation: added gpt-5-mini baseline and clarified that in the user study the humans could modify the code freely. However, as I describe above, some of the concerns are not fully resolved.

The authors have addressed the concerns with the sensitivity to the $\eta$ threshold, and also with their assistant potentially being more passive than the baselines.

**Reviewer Scores:**

- DgmQ: 4 -> 4 (main concern with the baselines and evaluation quality not fully-addressed)
- RDVP: 4 -> 4 (main concern with the connection between the method and empowerment not fully-addressed)
- Uf2z: 6 -> 6 (did not have major concerns, possibly with novelty and applications to other settings; unclear if would be willing to change the score)

---

### Decision · Program_Chairs · 2026-01-26

Reject